# Assessment of Parotid Gland Tumors by Means of Quantitative Multiparametric Ultrasound (mpUS)

**DOI:** 10.3390/diagnostics13010012

**Published:** 2022-12-21

**Authors:** Katharina Margherita Wakonig, Markus Herbert Lerchbaumer, Steffen Dommerich, Heidi Olze, Bernd Hamm, Thomas Fischer, Philipp Arens

**Affiliations:** 1Department of Otorhinolaryngology, Charité-Universitätsmedizin Berlin, Corporate Member of Freie Universität Berlin, Humboldt-Universität zu Berlin, and Berlin Institute of Health, Campus Virchow Klinikum and Campus Charité Mitte, Charitéplatz 1, 10117 Berlin, Germany; 2Department of Radiology, Charité-Universitätsmedizin Berlin, Corporate Member of Freie Universität Berlin, Humboldt-Universität zu Berlin, and Berlin Institute of Health, Charitéplatz 1, 10117 Berlin, Germany

**Keywords:** parotid gland tumors, multiparametric ultrasound, shear-wave elastography, CEUS

## Abstract

Objective: The preoperative diagnostical differentiation of parotid gland tumor (PGT) is not always simple due to several different entities. B-mode-ultrasound (US) remains the imaging modality of choice, while histopathology serves as the gold standard for finalizing the diagnosis. We aimed to evaluate the use of multiparametric US (mpUS) in the assessment of PGT. Methods: We included 97 PGTs from 96 patients. A standardized mpUS protocol using B-mode-US, shear-wave elastography (SWE), and standardized contrast-enhanced ultrasound (CEUS) was performed prior to surgical intervention. SWE was assessed by real-time measurement conducting a minimum of five measurements, while quantitative CEUS parameters were assessed with a post-processing perfusion software. Results: SWE allowed differentiation between benign PGT (Warthin’s Tumor (WT) paired with lymph nodes (LN) and pleomorphic adenoma (PA)), and WT and LN were softer compared to PA. WT showed lower velocities than squamous cell carcinoma (SCC): the most common malignant PGT. CEUS parameters showed significant group differences between WT and PA, WT and malignant lesions, WT and SCC, WT paired with LN versus PA, and WT paired with LN versus SCC. Conclusion: MpUS seems to be beneficial in the assessment of PGT characterization, with benign PGT appearing to be softer in SWE than tumors with malignant tendencies. The quantitative CEUS parameter shows higher perfusion in WT than in PA, and malignant PGTs are less vascularized than WTs.

## 1. Introduction

Ultrasound is the imaging modality of choice for parotid gland tumors (PGT) due to the easy access, fast availability, and the gland’s superficial location [1,2]. Most of the PGTs are benign, and only about 20% of all PGTs are known to be malignant [3], representing only 3% of head and neck cancers [4]. While primary salivary gland tumors are very rare and display a variety of histological entities [3], the metastasis of different skin cancers is more likely to be the cause of PGTs, with metastasis of cutaneous squamous cell carcinoma (SCC) being the most frequent malignant one [5,6,7]. The most common benign PGTs are pleomorphic adenomas (PA) and Warthin’s tumors (WT) [4]. While WT are exclusively benign, PAs have the tendency to transform into malignant tumors over time in approximately 7% of cases [8]. Even though US can visualize the size and localization and may suggest PGT entities by fulfilling certain US-B-mode criteria (such as shape, echogenicity, and tumor borders), a clear differentiation is not possible. Thus, surgery is still recommended to obtain a final diagnosis [1,9]. Parotid gland surgery bears various risks, such as damaging the facial nerve, postoperative hyperhidrosis (Frey’s syndrome), or salivary fistulas [10]. Even though US has become a valid tool in the preoperative diagnostics of PGT, entity prediction is not satisfying. The preoperative knowledge of a PGT entity determines the extent of surgery and, thus, should be improved [11]. In recent years, multiparametric ultrasound (mpUS) has revolutionized the US imaging technology primarily used in the evaluation of liver and kidney tumors. MpUS adds new applications to assess tissue composition by analyzing its stiffness by means of shear wave elastography (SWE) and visualize tumor microvascularization by adding a contrast-enhancing agent (contrast-enhanced ultrasound, CEUS) [12,13]. Our prospective cohort study aimed to assess the implementation of mpUS in the diagnostics of PGT by evaluating SWE and quantitative CEUS parameters.

## 2. Materials and Methods

### 2.1. Patient Selection

Overall, 96 patients met the inclusion criteria, which were defined as follows: (i) older than 18 years of age, (ii) with a parotid gland tumor of any size, and (iii) consenting histological confirmation and, therefore, surgery (partial or total parotidectomy) or US–guided core needle biopsies. In total, we included 97 PGTs because we performed surgery on both sides in one of the patients. Eligible patients were referred by the outpatient clinic of the otorhinolaryngological department of Charité—Universitätsmedizin Berlin (Campus Charité Mitte). Prior to surgery, mpUS was performed on every PGT. Please note that PGTs localized in the deep lobe were not included in this study because of US-imaging limitations in this region due to poor depth visualization and phase cancellation caused by the surrounding mandibular ramus. The study was approved by the institutional ethics committee of Charité—Universitätsmedizin Berlin (protocol code EA1/087/19, date of approval: 2 May 2019), and written informed consent was obtained from all participants according to the Declaration of Helsinki.

### 2.2. Imaging Protocol

A standardized US examination protocol was performed on every PGT with a single high-end US system with a 4–10 MHz multifrequency linear array transducer and a center frequency of 7 Mhz (Acuson Sequoia, Siemens Healthineers, Erlangen, Germany). For all PGTs, the short- and long-axis were measured, and SWE, as well as CEUS examination results, were obtained.

This was followed by a CEUS examination for which the participants were injected with a bolus of 2.4 mL of a second-generation contrast-enhancing agent (SonoVue^®^, Bracco Imaging, Milan, Italy) followed by an intravenous injection of 10 MT of sodium chloride 0.9%. At the beginning of the injection, a cine loop of 90 s was recorded to assess the tumor’s inflow and washout behavior. To avoid early microbubble rupturing examinations, each CEUS examination was performed with a low mechanical index (<0.1) and a state-of-the-art CEUS-specific protocol.

All examinations were performed by a trained otolaryngologist or an experienced radiologist. An overview of the imaging protocol is demonstrated in Figure 1 and Figure 2.

### 2.3. Shear Wave Elastography

All ultrasound-based SWE examinations were performed using a standardized protocol. For the examination of the parotid gland, patients were positioned in a relaxed supine position. Subjects were asked to stay as relaxed as possible. The US-SWE software (Virtual Touch™) allows for real-time measurement using acoustic radiation force impulse (ARFI) imaging technology for the measurement of shear wave speed (SWS). The quantitative measurement is expressed as the shear wave velocity (SWE) in meters per second (m/s). For each tumor lesion, we performed five consecutive SWE measurements using the 2D SWE approach. To acquire values of PGT stiffness, a circular region of interest (ROI) was placed on the tumor depicted by the dual-image mode (with corresponding B-Mode US image) for the controlled assessment of each lesion. The mean of five measurements and the corresponding standard deviation (SD) of the five consecutive measurements was calculated.

### 2.4. Perfusion Analysis

To quantify CEUS perfusion imaging in uncompressed DICOM files, we used the post-processing tool VueBox^®^ (Bracco Suisse SA-Software Applications, Geneva, Suisse). Tumor borders were manually marked to set an ROI, which stayed in place for the whole cine loop duration. In case of slight transducer movement, the implemented motion correction was used. The mean blood flow velocity was analyzed by time-related parameters, and the relative blood volume was measured by intensity parameters. These parameters and their corresponding areas under the curve (AUC) can assess relative blood flow independently from the microbubble inflow due to a linear relationship between the CEUS agent and echo-signal intensity. To minimize the risk of inaccurate inflow times due to the participants’ different cardiac outputs, the appearance of the first microbubble on the screen was used as a starting point for all time-related parameters. The obtained time-related parameters included rise time (RT), time to peak (TTP), mean transit time interval (mTTI), and fall time (FT) in seconds (s). The intensity-related parameters obtained included the median linearized signal of a region (meanLin), wash-in- and wash-out rate (WiR, WoR), wash-in and wash-out area under the curve as well as both the combined (WiAUC, WoAUC, and WiWoAUC) and wash-in perfusion index (WiPI) in arbitrary units as well as the peak-enhancement (PE) in decibel [14].

### 2.5. Statistical Analysis

Normal distribution was tested using the Kolmogorov–Smirnov test and was followed by the application of a dependent t-test. Variables following a normal distribution are reported as the mean and their associated standard deviation. Categorical variables were compared using Student’s *t*-test or Chi2-test, as appropriate. Subgroups were compared using the Mann–Whitney-U-test. A two-sided significance level of α = 0.05 was defined as appropriate to indicate statistical significance. All statistical analyses were performed using the SPSS software (IBM Corp. Released 2016. IBM SPSS Statistics for Windows, Version 27.0. Armonk, NY, USA: IBM Corp.).

## 3. Results

### 3.1. Study Cohort

Out of 96 patients, 36 were female (37.5%), and 60 were male (62.5%), with a general mean age of 62.4 years (SD ± 16.9 years). A total of 73.2% of the PGTs were histopathologically confirmed as benign (71/97), with the Warthin’s tumor being the most common entity (n = 38, 53.5% of benign PGT) followed by 21 pleomorphic adenomas (29.6% of benign PGT). Other benign PGTs were seven lymph nodes (LN) (9.9%), one intraglandular cyst, one oncocytoma, and three scars. Malignancy was depicted in 26.8% of all PGTs (26/97), showing 12 squamous cell carcinoma, four adenocarcinomas, four lymphomas, three malignant melanomas, one adenoid cystic carcinoma, one mucoepidermoid carcinoma, and one Merkel cell carcinoma. Detailed information can be found in Table 1.

### 3.2. Significant mpUS Parameters

The differences between PGT could be shown by SWE-velocity as well as by certain CEUS parameters. A general overview of significant mpUS parameters can be seen in Table 2. All PGT showed significant group differences in the intensity parameters of WiAUC, WoAUC, and WiWoAUC (see Figure 3 and Figure 4).

### 3.3. Shear Wave Elastography

WT + LN showed a significantly lower shear wave velocity (*p* = 0.012) than PA. A significant group difference (*p* < 0.001) was also found between WT + PA and LN with a higher SWE in WT and PA. Between WT and SCC, group differences were found (*p* = 0.046), with SCC demonstrating a higher SWE than WT. Further differences were found between WT and PA + SCC (*p* = 0.019), with WT showing a lower mean SWE. No differences were found between the benign and malign tumors. Detailed information can be found in Table 2, Table 3 and Table 4.

### 3.4. CEUS

#### 3.4.1. WT versus PA

Significant group differences favoring higher intensities in WT than in PA were found in MeanLin (*p* = 0.006), PE (*p* = 0.003), WiAUC (*p* = 0.005), WiR (*p* = 0.003), WiPI (*p* = 0.003), WoAUC (*p* = 0.006), WiWoAUC (*p* = 0.003), and WoR (*p* = 0.01). PA showed a longer mTTI (*p* = 0.015) and a longer FT than WT (*p* = 0.037). Please see Table 3 for more detailed information.

#### 3.4.2. WT versus Malignant Tumors (MT)

Significant differences between WT and MT were found in the intensity parameters MeanLin (*p* =0.008), PE (*p* = 0.008), WiAUC (*p* = 0.005), WiR (*p* = 0.01), WiPi (*p* = 0.009), WoAUC (*p* = 0.004), WiWoAUC (*p* = 0.004), and in the WoR (*p* = 0.006) with MT showing lower intensities by comparison. Please see Table 3 for more detailed information.

#### 3.4.3. WT versus SCC

Intensity parameters with significant differences between WT and SCC showed lower intensities for SCC in WiAUC (*p* = 0.047), WoAUC (*p* = 0.031), and in WiWoAUC (*p* = 0.033). No significant differences were found in the time parameters or remaining intensity parameters. Please see Table 3 for more detailed information.

#### 3.4.4. WT versus PA + SCC

The intensity parameter differences showing significantly higher values for WT compared to PA + SCC were MeanLin (*p* = 0.003), PE (*p* = 0.002), WiAUC (*p* = 0.002), WiR (*p* = 0.002), WoAUC (*p* = 0.002), WiWoAUC (*p* < 0.001), and WOR (*p* < 0.001).

The only intensity parameter in favor of PA paired with SCC was WiPi (*p* = 0.002). The only time parameter showing a significant group difference revealed a longer fall time (*p* = 0.014) and a longer mTTI (*p* = 0.03) in PA + SCC compared to WT. Please see Table 3 for more detailed information.

#### 3.4.5. WT + LN versus PA

Significant group differences in intensity parameters favoring WT + LN were MeanLin (*p* = 0.008), PE (*p* = 0.004), WiAUC (*p* = 0.008), WiR (*p* = 0.004), WiPI (*p* = 0.004), WoAUC (*p* = 0.01), WiWoAUC (*p* = 0.006), and WoR (*p* < 0.001). The only significantly differing time-related parameter was mTTI with a longer duration in PA (*p* = 0.019). Please see Table 4 for more detailed information.

#### 3.4.6. WT + LN versus Malignant Tumors (MT)

MeanLin (*p* = 0.012), PE (*p* = 0.012), WiAUC (*p* = 0.009), WiR (*p* = 0.013), WiPI (*p* = 0.012), WoAUC (*p* = 0.007), WiWoAUC (*p* = 0.007), and WoR (*p* = 0.01) showed significant group differences, all depicting higher intensities in WT paired with LN than in MT. Please see Table 4 for more detailed information.

## 4. Discussion

The present study findings in 96 consecutive patients with an overall 97 PGT can be summarized as follows: (i) SWE-velocity was significantly higher in PA and WT compared to LN and higher in PA compared to WT and LN, (ii) SWE-velocity was significantly higher in SCC than in WT, (iii) CEUS-analysis showed significant differences in the intensity- and time-related parameters between WT and PA, and (iv) malignant PGTs show a lower CEUS intensity (see Figure 1, Figure 2, Figure 3, Figure 4 and Figure 5).

Overall, the present study cohort’s characteristics match with the current literature by showing that 3/4 of PGTs were benign, with WT and PA being the most common entities. SCC was detected as the most common malignant entity.

Our results implicate an additional diagnostic gain through the multiparametric ultrasound of PGT. Even though the non-invasive SWE did not allow for the differentiation between WT and PA, it showed differences between benign LN and the two most common benign tumors combined, the latter showing a higher SWE velocity than LN. Previous work has demonstrated that SWE-velocity is proportional to tissue stiffness, indicating that shear waves travel faster in stiff tissue [15], and stiffness is, therefore, reflected as velocity. The low velocity in non-pathological LN is in line with our earlier findings, which showed that non-pathological lymph nodes of the neck are softer compared to malignant ones [16]. It has to be kept in mind that the parotid gland acts as a lymphatic drainage pathway in skin cancers of the forehead, around the ear, or the cheek [17]. In absence of PGT or skin cancer, the detection of an unsuspicious parotid LN is not a primary indication to perform surgery [18]. It is physiological to find up to eleven LN inside non-pathologically altered parotid glands, as post-mortem cadaveric studies have shown [19,20]. Lesion differentiation, with the additional help of SWE can aid the decision of whether surgery is needed, and unnecessary postsurgical complications could be avoided.

Another interesting result is the higher stiffness in PA compared to WT grouped with LN, which were paired based on the theory that WT originated from parotid LN and could, therefore, share the same characteristics [21]. This could also be explained by the composition of WT, which can be composed of cystic parts. Parotid gland cysts appear soft in mpUS and could maybe contribute to the softer appearance of these PGTs [22]. This could help to favor a watch-and-scan strategy, especially in patients with underlying medical conditions who have an elevated anesthesiologic risk. No significant group differences could be found between benign and malign PGT, which is in line with previous studies [22,23] and could be explained by the overall low number of parotid malignancies: a fact that could be owed to their rare occurrence and heterogeneity. Taking a closer look at SCC as the most common malignant PGT, we found that they showed a higher stiffness compared to WT, which was confirmed when SCCs were paired with Pas. Parotid SCCs are most commonly metastases of cutaneous SCCs [24,25]. The tissue of nodal SCC metastases has been proven to be rather stiff [16]. Moreover, PA showed a higher stiffness than the combination of WT and physiological LN. This could potentially indicate that WT and physiological LN, which are purely benign, appear rather soft, while SCC, as malignant PGT and PA, with a tendency of malignant transformation is stiffer.

While we were not able to differentiate between WT and PA by using SWE, quantitative CEUS parameters showed significant group differences. Additionally, the pairing of WT and physiological LN showed differences in these parameters compared to PA. While other study groups were only able to determine between PA and WT by TTP [26,27], we found differences in both the intensity parameters MeanLin, PE, WiAUC, WiR, WiPI, WoAUC, WiWoAUC, WoR, and time-related parameters such as FT and mTTI.

While WT (and LN) showed a stronger contrast enhancement, PA showed a longer mTTI, which can be interpreted as WT showing a higher perfusion than PA. This supports the findings presented by Klotz et al. [28].

The theory of higher perfusion is also supported by higher peak enhancement (PE) values in WT. High PE values are associated with an increase in perfusion or neovascularization as well as with a higher density of the microvessel structure, which results in a higher blood flow [29,30]. The increased AUC and PE values serve as a surrogate of higher contrast enhancement (e.g., higher microvessel density) in WT compared to PA, which was also recently shown by Welkoborsky et al. [31]. Higher WiR and WiPi values in WT compared to PA were also found by Klotz et al. [28].

A differentiation between WT (with and without pairing with physiological LN) and malignant tumors was possible with parametric CEUS, as well as a differentiation between WT and SCC by the intensity parameters. This led to the conclusion that malignant PGT showed lower contrast enhancement compared to WT, which may be a result of areas of necrosis within malignant tumors and, therefore, a lower vessel density. These results are not fully in line with previous findings, which showed contrary results [32] but can be explained by a high number of necrotic areas inside our cohort’s malignant PGT, which influence the determination of vascularization if they are inside the tumor’s ROI.

The ROI placement is a well-discussed limitation in CEUS perfusion analysis since there is no standardization. Some authors placed large ROIs in the whole tumor [33], while others put a couple of smaller ones throughout the whole tumor [31] or placed one in a normal parotid tissue on top [28,30,32]. Nevertheless, it is well known that the healthy PG is not clearly visualized, especially in larger PGTs, making perfusion assessment impossible. Another limitation is the PGT section that is analyzed; we only obtained a two-dimensional image during perfusion analysis. The acquired field of view depends on the placement of the US probe and is, therefore, dependent on the investigator’s choice. Since malignant PGTs are very rare and more inhomogeneous compared to benign PGTs [34], the effect of choosing an image section is a high potential bias.

Consistent with these observations, our results show that the intensity parameters WiAUC, WoAUC, and WiWoAUC seem to serve as the most important CEUS parameters in the differentiation of PGT, which summarize low intensities in malignant and higher intensities in benign PGT, as demonstrated in Figure 3 and Figure 4.

## 5. Conclusions

MpUS seems to be a good addition to B-mode US in the preoperative assessment of PGT entities and can easily be implemented in preoperative US routines. SWE indicates softer tissue in purely benign PGT and stiffer tissue in malignant tumors. CEUS seems to show higher perfusion in WT than in PA, and malignant PGT seems to be less vascularized than in WT. All PGT showed significant intensity differences in WiAUC, WoAUC, and WiWoAUC, which, therefore, seem to be the most important CEUS parameters in the differentiation of PGT. Further studies with a higher number of malignant PGTs should validate this method. The development of a standardized ROI placement for perfusion analysis should be considered.

## Figures and Tables

**Figure 1 diagnostics-13-00012-f001:**
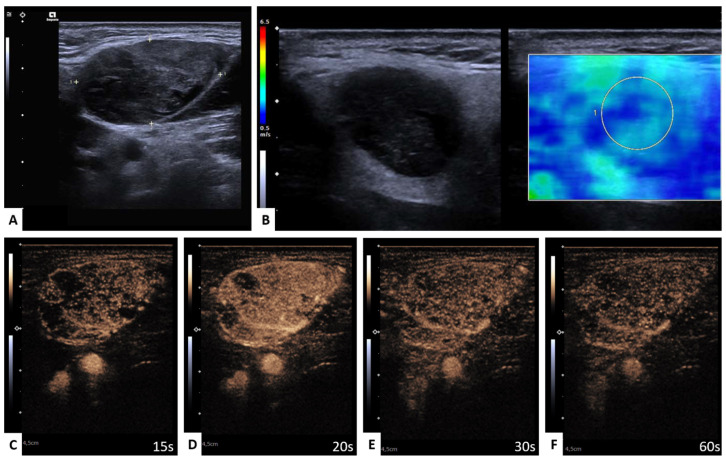
Benign parotid gland lesion; Warthin tumor. Representative image of mpUS workflow. (**A**) Long-axis and short-axis diameter measured in dual mode, (**B**) SWE measured by circular ROI revealed low tissue stiffness with a color-coded elastography scale from blue (soft) to red (hard) and the minimum and maximum velocity given in m/s on the left side, (**C**) CEUS image 15 s, (**D**) 20 s, (**E**) 30 s, and (**F**) 60 s after contrast injection with regular enhancement. Please note that decimal signs on the images are depicted as commas instead of period signs due to the German ultrasound software preset.

**Figure 2 diagnostics-13-00012-f002:**
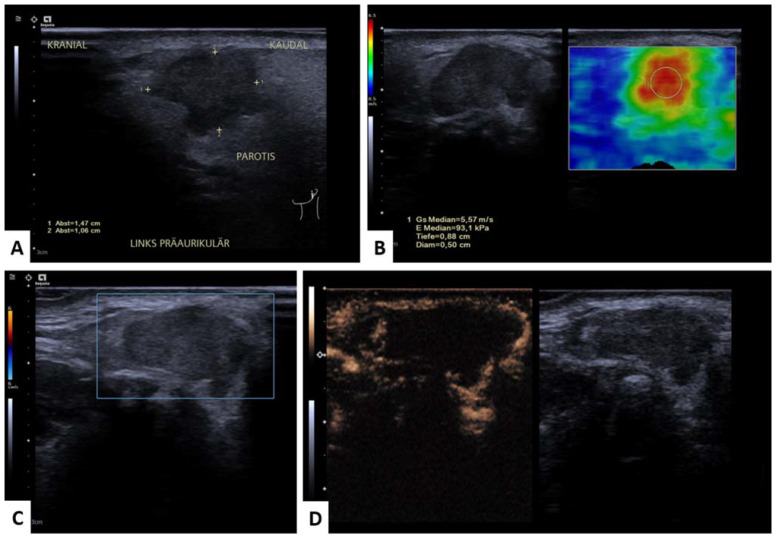
Malignant parotid gland lesion (squamous cell carcinoma metastasis). Representative image of mpUS workflow. (**A**) Long-axis and short-axis diameter measured in B-mode with german labeling of the image’s localization, (**B**) SWE measured by circular ROI revealed high tissue stiffness with a color-coded elastography scale from blue (soft) to red (hard) and the minimum and maximum velocity given in m/s on the left side. The ROI’s stiffness is depicted on every SWE-image’s left side in yellow. Please note that decimal signs on the images are depicted as commas instead of period signs due to the German ultrasound software preset. The symbol on the right side represents the position of the US-probe. (**C**) Color-coded Doppler sonography depicting no vascularization, (**D**) 60 s after contrast injection with avascular center (necrosis).

**Figure 3 diagnostics-13-00012-f003:**
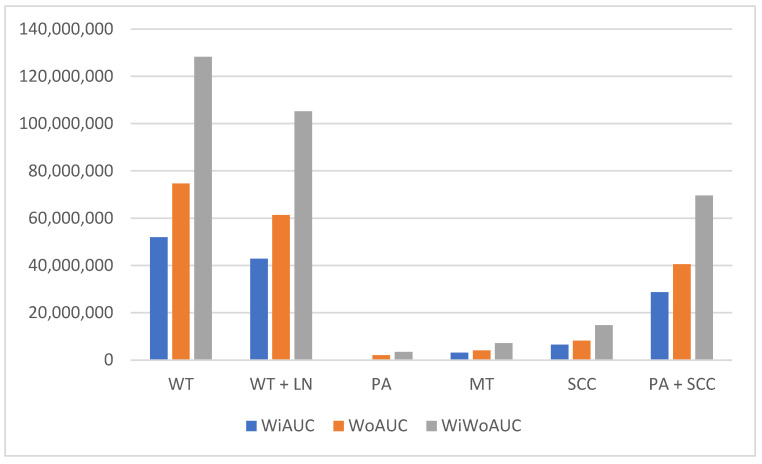
Bar graph depicting the significant differences between parotid gland tumors in the CEUS intensity parameters WiAUC (wash-in area under the curve), WoAUC (wash-out area under the curve), WiWoAUC (wash-in/wash-out area under the curve). X-axis: different parotid gland tumors WT (Warthin’s tumor); WT + LN (Warthin’s tumors + lymph nodes); PA (pleomorphic adenoma); MT (malignant tumors); SCC (squamous cell carcinoma); PA + SCC (pleomorphic adenoma + squamous cell carcinoma). Y-axis: mean intensity values of WiAUC, WoAUC, and WiWoAUC given in arbitrary units [a.u].

**Figure 4 diagnostics-13-00012-f004:**
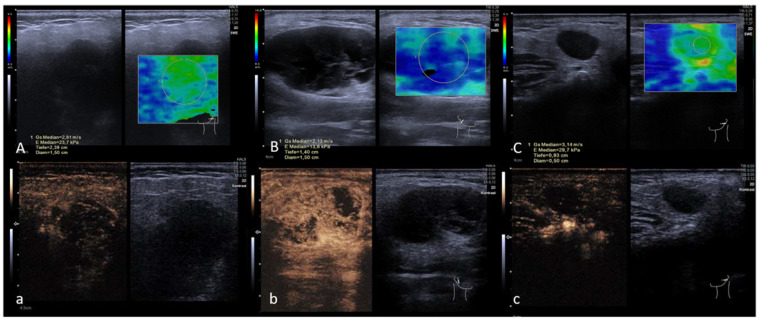
MpUS-visualization of the three most common entities: SWE (**A**) and CEUS (**a**) of pleomorphic adenoma (PA), SWE (**B**) and CEUS (**b**) of Warthin’s Tumor (WT) and SWE (**C**) and CEUS (**c**) of squamous cell carcinoma (SCC). CEUS images were taken 50 s after SonoVue^®^ injection with WT and were more vascularized compared to PA and SCC at the point of measurement and PA and SCC showed higher SWE-velocity compared to WT. SWE is measured by a circular ROI revealing tissue stiffness with a color-coded elastography scale from blue (soft) to red (hard) and the minimum and maximum velocity given in m/s on the left side. The ROI’s stiffness is depicted on every SWE-image’s left side in yellow. Please note that decimal signs on the images are depicted as commas instead of period signs due to the German ultrasound software preset. The symbol on the right bottom of the images represents the position of the US-probe.

**Figure 5 diagnostics-13-00012-f005:**
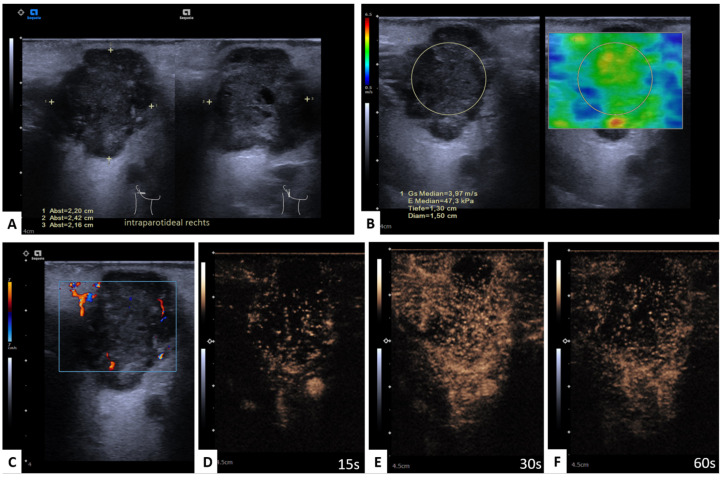
Malignant parotid gland lesion (squamous cell carcinoma metastasis). Representative image of mpUS workflow. (**A**) Long-axis and short-axis diameter measured in dual mode, (**B**) SWE measured by circular ROI revealed mediocre tissue stiffness with a color-coded elastography scale from blue (soft) to red (hard) and the minimum and maximum velocity given in m/s on the left side. The ROI’s stiffness is depicted on every SWE-image’s left side in yellow. Please note that decimal signs on the images are depicted as commas instead of period signs due to the German ultrasound software preset. The symbol on the right side represents the position of the US-probe. (**C**) Color-coded Doppler US depicting irregular vascularization, (**D**) CEUS image 15 s, (**E**) 30 s, and (**F**) 60 s after contrast injection with irregular enhancement.

**Table 1 diagnostics-13-00012-t001:** Study population. Continuous variables are given as mean (SD), categorical variables are given as absolute/total numbers (n/N), and percentages are in brackets. Abbrevations: PGT denotes parotid gland tumors.

Patients Female–Male Mean age	96 36/96 (37.5%)–60/96 (62.5%) 62.4 Y (±16.9 Y)
Benign PGT	71/97 (73.2%)
Pleomorphic Adenoma	21/71
Warthin’s Tumor	38/71
Lymph Node	7/71
Cyst	1/71
Oncocytoma	1/71
Other	3/71
Malignant PGT	26/97 (26.8%)
Squamous Cell Carcinoma	12/26
Adenocarcinoma	4/26
Lymphoma	4/26
Malignant Melanoma	3/26
Adenoid Cystic Carcinoma	1/26
Mucoepidermoid Carcinoma	1/26
Merkel Cell Carcinoma	1/26

**Table 2 diagnostics-13-00012-t002:** Main results: overview of significant SWE and CEUS results, “−“ denotes no signficant and “+” denotes significant SWE-results. Abbreviations: SWE denotes shear wave elastography; CEUS—contrast enhancing ultrasound; PGT—parotid gland tumors; PE—peak enhancement; WiAUC—wash-in area under the curve; RT—rise time; mTTI—mean transit time interval; TTP—time to peak; WiR—wash-in rate; WiPi—wash-in perfusion index; WoAUC—wash-out area under the curve; WiWoAUC—wash-in/wash-out area under the curve; FT—fall time; WoR—wash-out rate; vs.—versus.

Comparative PGT	SWE Significance	CEUS Significance
WT vs. PA	−	MeanLin, PE, WiAUC, WiR, WiPI, WoAUC, WiWoAUC, WoR, mTTI, FT
WT vs. MT	−	MeanLin, PE, WiAUC, WiR, WiPi, WoAUC, WiWoAUC, WoR
WT vs. SCC	+	WiAUC, WoAUC, WiWoAUC
WT vs. PA + SCC	+	MeanLin, PE, WiAUC, WiR, WoAUC, WiWoAUC, WOR
WT *+* LN vs. PA	+	MeanLin, PE, WiAUC, WiR, WiPI, WoAUC, WiWoAUC, WoR, mTTI
WT + LN vs. MT	−	MeanLin, PE, WiAUC, WiR, WiPI, WoAUC, WiWoAUC, WoR

**Table 3 diagnostics-13-00012-t003:** Results of Warthin’s tumor compared to different entities. All metric variables are given as the median with corresponding standard deviation; significant *p*-values are pointed out in bold. Abbreviations: SWE denotes shear wave elastography; CEUS—contrast enhancing ultrasound; vs.—versus; WT—Warthin’s tumor; PA—Pleomorphic Adenoma; MT—malignant tumor; SCC—squamous cell carcinoma; *p*—*p*-value; PE—peak enhancement; WiAUC—wash-in area under the curve; RT—rise time; mTTI—mean transit time interval; TTP—time to peak; WiR—wash-in rate; WiPi—wash-in perfusion index; WoAUC—wash-out area under the curve; WiWoAUC—wash-in/wash-out area under the curve; FT—fall time; WoR—wash-out rate; s—seconds; n—number; [a.u]—arbitrary units.

	WT	vs. PA	*p*	vs. MT	*p*	vs. SCC	*p*	vs. PA + SCC	*p*
**SWE**	n = 38	n = 21		n = 26		n = 12		n = 33	
**SWE (m/s)**	2.5 ± 0.92	2.94 ± 1.03	0.069	2.63 ± 1	0.339	**3.09 ± 1.04**	**0.046**	**2.73 ± 0.99**	**0.019**
**CEUS**	n = 33	n = 20		n = 23		n = 10		n = 30	
**meanLin** **[a.u]**	2,006,078.77 ± 6,022,658.47	59,388.64 ± 175,556.44	**0.006**	121,719.86 ± 514,813.21	**0.008**	247,028.91 ± 780,708.07	0.054	1,108,867.64 ± 4,441,027.18	**0.003**
**PE** **[a.u]**	19,072,765.52 ± 61,324,843.66	540,672.11 ± 1,592,891.29	**0.003**	1,468,390.72 ± 6,490,545.51	**0.008**	3,115,717.74 ± 9,851,282.77	0.062	10,656,696.54 ± 45,119,544.60	**0.002**
**WiAUC** **[a.u]**	51,905,344.84 ± 159,318,110.57	1,308,043.56 ± 3,927,811.86	**0.005**	3,135,246.66 ± 13,496,189.56	**0.005**	6,472,313.56 ± 2,0461,410.99	**0.047**	28,631,117.25 ± 117,364,223.12	**0.002**
**RT (s)**	6.22 ± 2.59	6.51 ± 3.02	0.714	6.27 ±1.77	0.459	6.52 ± 1.63	0.239	6.36 ± 2.58	0.401
**mTTI (s)**	52.26 ± 63.38	76.74 ± 55.22	**0.015**	66.08 ± 68.81	0.6	66.47 ± 59.44	0.472	62.29 ± 60.36	**0.03**
**TTP (s)**	10.10 ± 2.95	9.01 ± 3.58	0.102	9.34 ± 2.08	0.566	9.83 ± 1.56	0.863	9.71 ± 3	0.253
**WiR** **[a.u]**	582,2087.69 ± 19,288,365.48	190,810.89 ± 557,466.92	**0.003**	543,573.01 ± 2,446,514.59	**0.010**	1,174,780.24 ± 3,714,618.56	0.062	3,296,712.92 ± 14,189,853.33	**0.002**
**WiPI** **[a.u]**	11,705,571.65 ± 37,461,700.07	336,761.33 ± 992,297.43	**0.003**	896,490.65 ± 3,958,811.46	**0.009**	1,900,363.97 ± 6,008,529.95	0.062	6,540,043.35 ± 27,567,268.56	**0.002**
**WoAUC** **[a.u]**	74,698,989.56 ± 218,145,521.46	2,057,520.60 ± 6,158,753.67	**0.006**	4,023,168.72 ± 17,103,206.84	**0.004**	8,198,558.42 ± 25,914,563.94	**0.031**	40,540,381.65 ± 159,886,884.39	**0.002**
**WiWoAUC** **[a.u]**	128,226,364.53 ± 379,440,971.04	3,362,932.11 ± 10,087,063.36	**0.003**	7,158,414.81 ± 30,598,900.51	**0.004**	14,670,870.98 ± 46,375,971.76	**0.033**	69,632,435.75 ± 277,944,250.82	**<0.001**
**FT (s)**	11.96 ± 5.84	16.15 ± 10.39	**0.037**	15.10 ± 8.90	**0.076**	14.38 ± 5.24	0.059	13.70 ± 7.66	**0.014**
**WoR** **[a.u]**	3,818,240.26 ± 13,233,643.32	95,771.36 ± 278,806.23	**0.01**	371,129.02 ± 1,685,672.00	**0.006**	809,402.23 ± 2,559,420.97	0.063	213,2147.39 ± 9,650,318.62	**<0.001**

**Table 4 diagnostics-13-00012-t004:** Results of Warthin’s tumors paired with lymph nodes compared to different entities. All metric variables are given as the median with corresponding standard deviation; significant *p*-values are pointed out in bold. Abbreviations: SWE denotes shear wave elastography; CEUS—contrast enhancing ultrasound; vs.—versus; WT—Warthin’s tumor; PA—Pleomorphic Adenoma; MT—malignant tumor; LN—lymph node; *p*—*p*-value; PE—peak enhancement; WiAUC—wash-in area under the curve; RT—rise time; mTTI—mean transit time interval; TTP—time to peak; WiR—wash-in rate; WiPi—wash-in perfusion index; WoAUC—wash-out area under the curve; WiWoAUC—wash-in/wash-out area under the curve; FT—fall time; WoR—wash-out rate; s—seconds; n—number; [a.u]—arbitrary units.

	WT + LN	vs. PA	*p*	vs. MT	*p*
**SWE**	n = 45	n = 21		n = 26	
**SWE (m/s)**	2.33 ± 0.95	2.94 ± 1.03	**0.012**	2.63 ± 1	0.086
**CEUS**	n = 40	n = 20		n = 23	
**MeanLin** **[a.u]**	1,655,586.27 ± 5,509,624.98	59,388.64 ± 175,556.44	**0.008**	121,719.86 ± 514,813.21	**0.012**
**PE** **[a.u]**	15,740,147.40 ± 56,030,655.45	540,672.11 ± 1,592,891.29	**0.004**	1,468,390.72 ± 6,490,545.51	**0.012**
**WiAUC** **[a.u]**	42,835,826.26 ± 145,685,272.30	13,08043.56 ± 3,927,811.86	**0.008**	3,135,246.66 ± 13,496,189.56	**0.009**
**RT (s)**	6.13 ± 2.39	6.51 ± 3.02	0.649	6.27 ±1.77	0.392
**mTTI (s)**	53.13 ± 59.39	76.74 ± 55.22	**0.019**	66.08 ± 68.81	0.732
**TTP (s)**	9.80 ± 2.79	9.01 ± 3.58	0.158	9.34 ± 2.08	0.92
**WiR** **[a.u]**	4,804,810.78 ± 17,614,431.92	190,810.89 ± 557,466.92	**0.004**	543,573.01 ± 2,446,514.59	**0.013**
**WiPI** **[a.u]**	9,660,280.10 ± 34,230,356.76	336,761.33 ± 992,297.43	**0.004**	896,490.65 ± 3,958,811.46	**0.012**
**WoAUC** **[a.u]**	61,313,661.69 ± 199,153,143.51	2,057,520.60 ± 6,158,753.67	**0.010**	4,023,168.72 ± 17,103,206.84	**0.007**
**WiWoAUC** **[a.u]**	105,247,832.72 ± 346,310,544.32	3,362,932.11 ± 10,087,063.36	**0.006**	7,158,414.81 ± 30,598,900.51	**0.007**
**FT (s)**	12.08 ± 5.48	16.15 ± 10.39	0.053	15.10 ± 8.90	0.107
**WoR** **[a.u]**	313,3745.45 ± 1,204,4372.37	95,771.36 ± 278,806.23	**<0.001**	371,129.02 ± 1,685,672.00	**0.01**

## Data Availability

The datasets generated and analyzed during the current study are not publicly available but are available from the corresponding author on reasonable request.

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
