# Peer review of "Assessment of Parotid Gland Tumors by Means of Quantitative Multiparametric Ultrasound (mpUS)"

_diagnostics, 2022, doi:10.3390/diagnostics13010012_

Round 1

Reviewer 1 Report

This manuscript describes the feasibility of multiparametric ultrasound (mpUS) as an assessment method for parotid gland tumors in a case-series fashion, which has significant potential to provide information not only about mass structures but also histopathological implications which can be associated with its malignant features.

Major concerns:
1. The study design and the hypothesis are unclear. Is the study design a cohort or retrospective? Is the main purpose of this study to compare classical US and mpUS, or to compare SWE and CEUS? These should be led logically from the previous evidence mentioned in the introduction.
2. All the figures and the tables seem to be lax and need to contain relevant information for interested readers to better understand the significance.
i. All the tables lack measurement units in each variable. Moreover, unfortunately, the numerical data shown imply less awareness for significant digits.
ii. It may be a good idea to show the other examples of mpUS images as representative types of parotid tumors for interested readers.

Minor points:
3. The tables should have redundancies eliminated. For example, males and females can be shown in a single row.
4. I believe that the author can present the most significant findings in a visualized manner, such as a bar graph or pi chart for readers to understand the feasibility of mpUS.

I think that the authors could reconsider these points to make the manuscript relevant and show how it can be beneficial for interested readers.

I hope you kindly consider my suggestions.

Reviewer 2 Report

The original article entitled “Assessment of Parotid Gland Tumors (PGTs) by means of Quantitative Multiparametric Ultrasound (mpUS)” analyzed 97 PGT of 96 patients, the patients received standardized mpUS protocol using B-mode-US, shear-wave elastography (SWE) and standardized contrast-enhanced ultrasound (CEUS) was performed prior to surgical intervention.

The followings are my comments:

1. The application and combination of ultrasonic tools is of clinical importance, and there are few well-structured articles focus on parotid gland. This is an interesting topic that will attract the surgeons’ attention.

2. Some patient information not provided:

  (1) Size, tumors that are too small or too large are not candidates for current method. Authors are required to describe patient selection and share experience regarding the appropriate scale of assessment.

  (2) Depth and location, for differences in deep/superficial lobe, preauricular/tail of parotid, etc., whether it affects the current results?

  (3) Composition, whether it affects the evaluation in non-solid tumors, especially in areas containing cysts, and whether cysts need to be aspirated first?

3. The parameters of CEUS show statistically significant for tumor comparisons, and the authors may provide your suggestion as to which parameters are most essential for differentiating tumors.

4. (1) May abbreviate MT for “malignant tumors”, or use ML for “malignant lesions.” (2) Currently description is “WT versus PA and SCC”, may change to “WT versus PA+SCC” would be easier for readers to understand.

5. At present, there are too many tables but the forms are similar, it is suggested that combine 2-3 tables into one or change to horizontal layout. In addition, it is recommended to add a table, including: (1) comparative lesions (ex. WT vs. PA); (2) SWE significance (+ or -) (3) CEUS significance (WiAUC, WoAUC, WiWoAUC…) (4) Provide a short interpretation for the results. It will allow readers to better understand the results and discussions of this study.

This study provides complete clinical evaluation and data, and makes appropriate statistical analysis, and the manuscript were well-written. This article highly readable and clinically applicable, and should be valued by readers.
